# Improving Skin Hydration and Age-related Symptoms by Oral Administration of Wheat Glucosylceramides and Digalactosyl Diglycerides: A Human Clinical Study

**Valérie Bizot [1,\*], Enza Cestone [2], Angela Michelotti [2] and Vincenzo Nobile [2]** 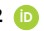

[1] Extraction Purification Innovation France–E.P.I. France, R&D department, 3 rue de Préaux, Villers sur Fère 02130, France

[2] Farcoderm Srl, Member of Complife Group, Via Mons Angelini 21, San Martino Siccomario, Pavia 27028, Italy; enza.cestone@complifegroup.com (E.C.); angela.michelotti@complifegroup.com (A.M.); vincenzo.nobile@complifegroup.com (V.N.)

\* Correspondence: v.bizot@epifrance.fr; Tel.: +33-323-822897

**Abstract:** Ceramides are known to play a key role in the skin's barrier function. An age-dependent decrease in ceramides content correlates with cutaneous clinical signs of dryness, loss of elasticity, and increased roughness. The present placebo-controlled clinical study aims to evaluate if an oral supplementation with glucosylceramides (GluCers) contained in a wheat polar lipids complex (WPLC) was able to improve such skin conditions. Sixty volunteers presenting dry and wrinkled skin were supplemented during 60 days with either a placebo or a WPLC extract in oil or powder form (1.7 mg GluCers and 11.5 mg of digalactosyldiglycerides (DGDG)). Skin parameters were evaluated at baseline and after 15, 30, and 60 days of supplementation. Oral intake of WPLC significantly increased skin hydration ($p < 0.001$), elasticity, and smoothness ($p < 0.001$), and decreased trans epidermal water loss (TEWL) ($p < 0.001$), roughness ($p < 0.001$), and wrinkledness ($p < 0.001$) in both WPLC groups compared to placebo. In both WPLC treated groups, all parameters were significantly improved in a time-dependent manner compared to baseline. In conclusion, this study demonstrates the positive effect of oral supplementation with GluCers on skin parameters and could reasonably reinforce the observations made on mice that orally-supplied sphingolipids can reach the skin.

**Keywords:** hydration; skin barrier function; anti-aging; glucosylceramides; food supplement; human clinical trial

## 1. Introduction

An important function of the skin is to provide an effective barrier against the loss of water and electrolytes. The properties of this permeability barrier are correlated with the role of *stratum corneum* (SC) lipid lamellae in forming the intercellular layer with a unique and very different composition from the lipids that compose biological membranes. Human SC lipids are composed of 50% ceramides, 25% cholesterol, and 15% free fatty acids (FFAs). All three components are required for skin integrity, especially the ceramides, which play a crucial role in bilayer system formation [1]. Ceramides form a complex and structurally heterogeneous group of sphingolipids; they are composed of a sphingoid base linked by an amide bond with a fatty acid (FA). Among them, acyl-ceramides (where linoleic acid is esterified on ω-hydroxy fatty acids) play an important role in the cohesion between the different intercellular lipid membranes and in water retention [2,3].

With aging, human skin becomes thinner, wrinkled, and loses some of its elasticity.

Dryness observed in aged skin correlates with an overall decrease of approximately 30% in total SC lipids [4]. In particular, it has been demonstrated that total SC ceramide content declines with age [5]. Additionally, it has been reported that age and season modify ceramides subclasses profiles in human SC. Indeed, the level of ceramide I and ceramide I linoleate (C18:2) decreases significantly between a young (26–29 years old) group and an older (57–60 years old) group [6,7].

Consequently, cosmetic research has focused for decades on finding evidence that topical application of ceramides and/or sphingolipids to the skin leads to improved moisture regulation, smoother and more elastic skin, and a general amelioration of the cutaneous barrier [8]. More recently, scientists have investigated the beneficial effects of oral supplementation with ceramides to improve dry skin, skin aspect, and associated discomforts. Animal studies have described ceramide bioavailability after ingestion. Dietary glycosylceramides were found to metabolize in rat small intestine: about half of the dietary substrate was absorbed, and was found in portal blood after hydrolysis by ceramidases in the gastrointestinal tract [9]. Even though a large proportion of ingested sphingolipids are excreted in the feces, animal studies reported that after oral intake radiolabeled ceramides are metabolized, absorbed and distributed to many tissues, including the skin. In rats, orally administered radiolabeled ceramides were shown to be delivered to the epidermis [10]. In another study on mice, after administration of $^{13}$C-labeled dihydroceramides, their metabolite $^{13}$C-labeled sphinganine was clearly detected in the skin, liver, skeletal muscle and synapse membrane in the brain [11]. It has also been shown that orally-administrated radiolabeled D2-sphingosine is transferred to the skin, from dermis to epidermis in an unchanged structural form, and further generates radiolabeled glucosylceramides and ceramides by in vivo biosynthesis in mice [12].

In addition, there is a slowly growing body of evidence from animal and human clinical studies that oral supplementation with ceramides may be beneficial for skin permeability barrier homeostasis [13–18] and parameters such as hydration and/or barrier function, elasticity, and recovery after induced disruption of barrier dysfunction.

The main dietary sphingolipid species that we ingest from cereals and plants is represented by glucosylceramides [12,19–24]. Cereal sphingolipids intake represents an average of 76 mg/day [24], corresponding to more than 25% of daily total sphingolipids.

Based on this interesting background, a proprietary wheat polar lipid complex (WPLC) was developed and produced in purified forms: a concentrated powder (WPLC-P) with high polar lipids content (≥98%) and an oil form (WPLC-O) rich in polar lipids (≥45%). It contains sphingolipids, also called phytoceramides, that included glucosylceramides and a wheat natural emulsifier, digalactosyl diglycerides (DGDG). In a pilot human clinical study, we have demonstrated that oral intake of 20 mg/day of WPLC in powder form (WPLC-P) can improve skin's moisturization index, elasticity, and skin microrelief significantly after only 15 days. Indeed, skin smoothness was increased while skin roughness and microwrinkles were decreased (unpublished data).

Based on these primary data, we decided to test the efficacy of oral intake of purified WPLC (containing glucosylceramides and DGDG) on skin hydration, barrier function, and aged-related symptoms in a placebo-controlled, randomized, and double-blind clinical trial with healthy human volunteers with dry and wrinkled skin.

## 2. Materials and Methods

### 2.1. Trial Design

This was a monocentric double-blind, placebo-controlled, randomized study performed on sixty healthy Caucasian female subjects. All of the study procedures were carried out according to the World Medical Association's (WMA) Helsinki Declaration and its amendments (Ethical Principles for Medical research Involving Human Subjects, adopted by the 18th WMA General Assembly Helsinki, Finland, June 1964 and amendments). The study protocol and the informed consent form were

approved by an independent regional ethical committee (Code N 22-102009, Comitato Etico di Ricerca del Centro Analisi Monza S.p.A., Monza, Italy).

Information on the objectives and procedure of the trial, on dietary recommendations and on study benefits and risks were provided to the volunteers before their participation. The subjects were then asked for written informed consent before participating in the trial.

After a 5-day conditioning period, the volunteers were randomly assigned to three different intervention groups: placebo, WPLC-O (wheat polar lipids complex-oil form, 70 mg/day) and WPLC-P (wheat polar lipids complex-powder form, 30 mg/day). The conditioning period was used to allow all subjects to adapt their habits to the study requirements. During this period, all of the volunteers received a basal night and day cream (with no claimed effect on skin by topical application) in order to standardize topical application on the skin. The cream was applied by the volunteers to the face twice a day, in the morning and at night. Skin hydration was assessed at baseline and after 15, 30, and 60 days of supplementation with the tested products using Corneometer$^{®}$ CM 825 (Courage + Khazaka electronic GmbH, Cologne, Germany) and Tewameter$^{®}$ TM300 (Courage + Khazaka electronic GmbH, Cologne, Germany).

All of the analyses were centralized by Farcoderm Srl, Member of Complife Group, Via Mons Angelini 21, San Martino Siccomario 27028, Pavia, Italy.

No changes occurred in the methods after trial commencement.

*2.2. Participants*

2.2.1. Eligibility Criteria for Participants

Healthy women aged 30–60 years presenting dry skin (Corneometer$^{®}$ value < 50) and showing clinical signs of face skin aging related to photoaging (with medium photoaging signs, dry and devitalized skin, pale/greyish skin, early aging signs caused by slowing in cell activity) or mild-to moderate chrono-aging according to Fitzpatrick classification [25] were recruited. Corneometer$^{®}$ value for dry skin was set up based on "Information and Operating Instructions for the Corneometer$^{®}$ CM 825 Stand-alone" and with software version "CM825 alleine English 07/2007 DK" and on Farcoderm lab experience.

Exclusion criteria were an abnormal medical check-up, an obvious skin disease, an abnormal body weight (Body mass index: BMI < 19 and > 30 kg/m$^2$), a known history of lipid metabolism disorders, and intensively exposure to the sun (natural or artificial) for at least two months. Additional, exclusion criteria were pregnancy or intention to become pregnant, lactation, food allergy/intolerance, and participation in another similar study within the last two months prior to enrollment in the present study, using products containing moisturizing and/or anti-aging active ingredients taken orally or applied topically. The study further excluded subjects using tanning beds and undergoing dermatological (including peeling) or pharmacological (either local or systemic, i.e. corticoids, retinoids, vitamins) treatments within the last two months before the beginning of the study. Finally, all subjects were asked not to change their lifestyle, toilettes and dietary habits. Indeed, subjects unable/unwilling to comply with protocol requirements (including accordance not to use any other cosmetic product than the basal cream given at the beginning of the study) were not included in the study.

2.2.2. Settings and Locations

The study took place at Farcoderm Srl dermatological facilities in San Martino Siccomario (PV), Italy. Farcoderm Srl is an independent certified testing laboratory, collaborating with the University of Pavia, for in vitro and in vivo safety and efficacy assessment of cosmetics, food supplements, and medical devices.

*2.3. Interventions*

2.3.1. Experimental Wheat Polar Lipids Complex: Production and Characterization

A food-grade wheat polar lipids complex (WPLC) was extracted and purified according to a proprietary manufacturing process. Two grades were produced: an oil form (WPLC-O) and a concentrated powder form (WPLC-P). Briefly, for WPLC-O, this process consisted of successive water/ethanol extractions. After solid/liquid separation, the oil extract was concentrated under vacuum. WPLC-P was obtained by successive water/ethanol extractions, solid/liquid separation, and purification with acetone, followed by high-vacuum drying. Both forms of WPLC were extracted from selected wheat (*Triticum aestivum*, also named *vulgare* or *sativum*) endosperm flour.

WPLC was firstly characterized by mass spectrometry (MS). A direct electrospray/MS was performed for galactolipid characterization. Hydrolyzed WPLC was analyzed by gas chromatography/mass spectrometry (GC/MS) after derivatization and secondly by electrospray ionization/MS to assess the molecular mass of the substance and the filiation of fragments obtained by collision-activated dissociation (CAD) using an ESQUIRE (Bruker, Wissembourg, France) mass spectrometer with an ion trap. MS data on WPLC highlighted the presence of galactolipids, including digalactosyldiglycerides (DGDG) and glycosphingolipids, particularly glycosylceramides (Table 1).

**Table 1.** Lipid and fatty acids (FAs) composition identified and characterized by GC/MS.

| Constituent Structures | Weighted Molecular Mass |
|---|---|
| **Major GALACTOLIPIDS** | |
| Digalactosyl diglycerides (DGDG)<br>FAs composition:<br>    C18:2 | 940 g/mol |
| **Major GLYCOSYLCERAMIDES** | |
| Sphingoid bases:<br>    t 18:0 phytosphingosine (sphinganine)<br>    d 18:1 sphingosine (8 sphingenine)<br>FAs composition:<br>    C16 to C18 (saturated and unsaturated)<br>Sugar composition:<br>    Minimum 1 sugar | Average: 737 g/mol<br>(from 716 to 756 g/mol) |

Secondly, to obtain more details in WPLC sphingolipid characterization, an ultra-performant liquid chromatography–electrospray mass spectrometry/mass spectrometry (UPLC–ESI-MS/MS, Waters, Manchester, UK) study was performed using multiple reaction monitoring (MRM) mode, a method in which the eluate was continuously scanned for selected precursor–product ion pairs to enhance the sensitivity and specificity of the analysis of different classes of sphingolipids, as described in a recent publication [26]. Results were consistent with the first MS analysis, particularly for the identification of glycosylated phytoceramides. This UPLC–ESI-MS/MS study highlighted that WPLC contains more than 162 individual molecular species of ceramides, including glycosylceramides (data not shown). This high diversity in ceramides structure correlates with the presence of four sphingoid bases (t 18:0, t 18:1, d 18:0, d 18:1) and an amine linkage with a large diversity of FAs (from C16 to C26, in non-hydroxylated and hydroxylated forms). In the glycosphingolipids class, one species was clearly identified as glucosylceramide, where one molecule of glucose is esterified on the sphingoid bases of ceramides. At present, there is no routine technical solution to quantify all ceramide species. Thus, the more detailed analysis in WPLC has focused on the GluCers class, which have been previously defined as the principal markers of ceramides from plant sources [19–21]. Data are presented in Table 2.

**Table 2.** GluCers identified and characterized by UPLC-ESI-MS/MS.

| GluCers Structure | GluCers Molecular Mass |
|---|---|
| Sphingoid bases: t 18:0 = phytosphingosine (4-hydroxysphinganine) t 18:1 = hydroxy sphingosine (4-hydroxy-8-sphingenine) d 18:0 = dihydrosphingosine (sphinganine) d 18:1 = sphingosine (8-sphingenine) FAs composition: C16 to C26 (hydroxylated and non-hydroxylated forms, saturated and unsaturated forms) Sugar composition: Monoglycosylated with 1 molecule of glucose | Average: 760 g/mol (from 716 to 872 g/mol) |

High-field NMR spectra for proton ($^1$H) and carbon-13 ($^{13}$C) were obtained on a Bruker Avance DPX 500 spectrometer running at 500 MHz (Bruker, Wissembourg, France), after dissolution of WPLC in deuterated chloroform (CDCl3). The presence of ceramides is attested both by $^1$H and $^{13}$C-NMR (Figure 1).

Total esterified FA composition was quantified by GC following the standard methods references NF EN ISO 12966-2 and NF EN ISO 5508. Both forms of WPLC contained high levels of polyunsaturated fatty acids (>78%), including linoleic acid C18:2 (n−6) >60% for WPLC-O and >65% for WPLC-P.

Quantification of DGDG content was performed by an in-house validated HPLC method by reverse phase chromatography/ELSD using wheat DGDG as standard, from Larodan (ref: 59-1210-8).

Total sphingolipids content was quantified by an indirect internal method based on the following principle: one molecule of sphingolipids (or glycosphingolipids) contains one atom of nitrogen. Total nitrogen content was quantified by the Dumas method (NF EN ISO 16634-1). Nitrogen content coming from proteins and phospholipids was subtracted. Then, taking into account the average molecular weight of glucosylceramides characterized by MS (Table 1), calculation yielded an estimation of total sphingolipid content. More precisely, wheat GluCers markers in WPLC were quantified by normal phase high-performance liquid chromatography/evaporative light scattering detection (HPLC/ELSD) [22] using glucosylceramides from wheat as standard (Nacalaï ref NS170703).

Specifications of WPLC are reported in Table 3.

**Table 3.** WPLC active ingredients composition.

| WPLC grade | DGDG (HPLC Method) | Total Sphingolipids (Nitrogen Method) |
|---|---|---|
| WPLC-P | ≥40% | ≥50% |
| WPLC-O | ≥15% | ≥15% |

The content of wheat GluCers markers (HPLC) in the clinical tested batches of WPLC are as follows: 53 mg/g for WPLC-P and 24 mg/g for WPLC-O.

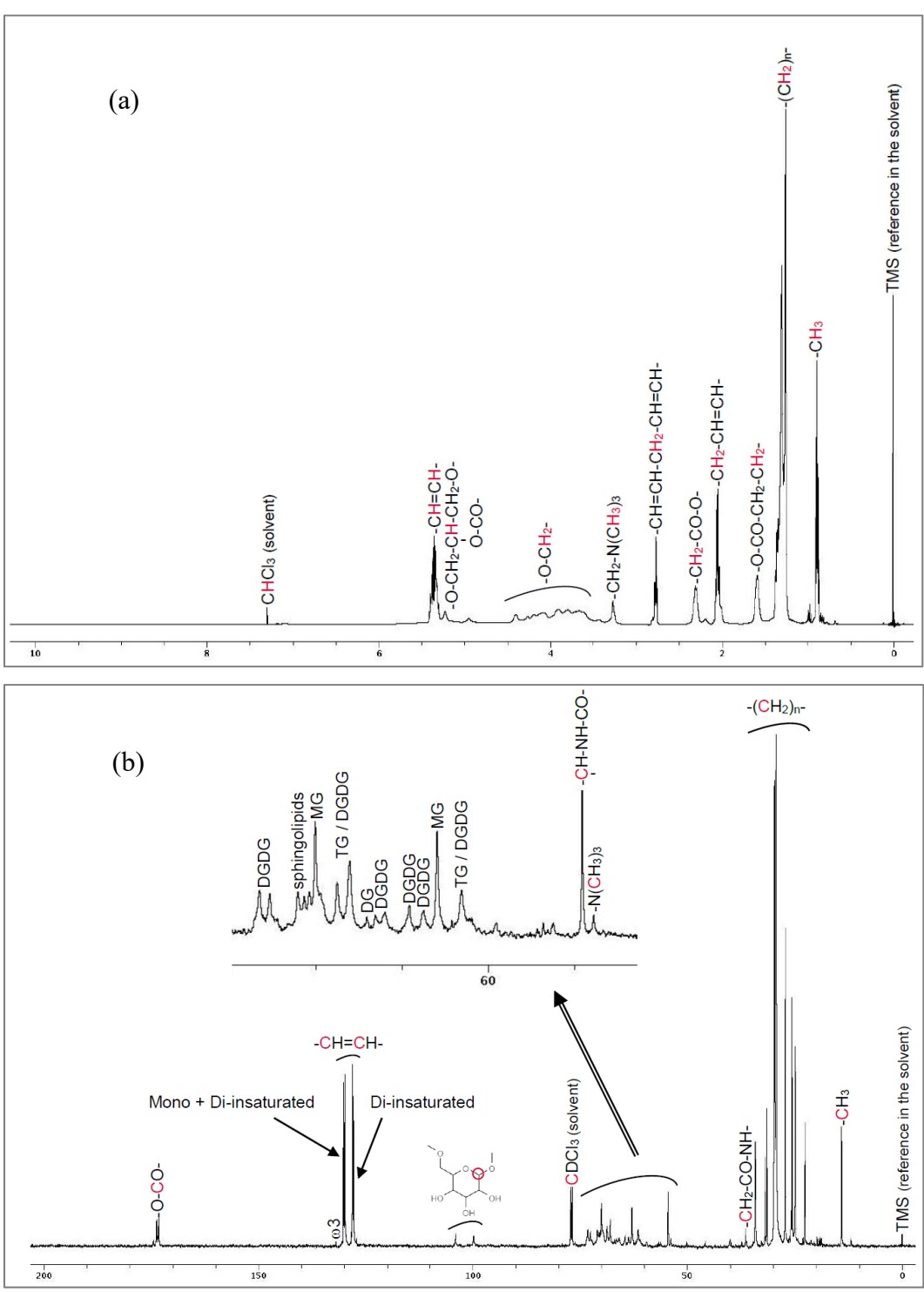

**Figure 1.** High field NMR spectra of WPLC; (**a**) $^1$H-NMR spectra and (**b**) $^{13}$C-NMR (TG: Triglycerides, DG: Diglycerides, MG: Monoglycerides).

### 2.3.2. Experimental Products: Description and Intake

During the 60-day experimental period, the daily oral intake was two capsules, at night before sleeping, containing either placebo (maltodextrin), WPLC-O extract (=70 mg/day) and WPLC-P extract (=30 mg/day). All capsules were identical in terms of size, color and odor and their composition is

presented in Table 4. Indeed, WPLC-P and WPLC-O groups were, respectively, supplemented with 1.6 mg/day and 1.7 mg/day of wheat glucosylceramides and 12 mg/day and 11 mg/day of DGDG.

**Table 4.** Qualitative and quantitative formula of test products (per capsule).

| Composition | Placebo (mg) | WPLC-O (mg) | WPLC-P (mg) |
|---|---|---|---|
| Maltodextrin | 215 | 180 | 200 |
| Dicalcium phosphate | 100 | 100 | 100 |
| Magnesium carbonate | 50 | 50 | 50 |
| Silice | 30 | 30 | 30 |
| WPLC-P | 0 | 0 | 15 |
| WPLC-O | 0 | 35 | 0 |
| Magnesium stearate | 5 | 5 | 5 |
| Total weight | 400 | 400 | 400 |

The volunteers had to record their daily food consumption (using a 33 item food questionnaire) over the first 2 weeks of the intervention period (day 1 to 7 and then day 8 to 14) to ensure they did not change their dietary habits during the study course.

*2.4. Outcomes*

The primary efficacy endpoint was the evaluation of skin hydration measured in 5 replicates in the cheeks by Corneometer® CM 825 at day 60.

Secondary efficacy endpoints included the assessment of:

- skin hydration at day 15 and 30 by Corneometer® CM 825;
- transepidermal water loss (TEWL, 1 replicate measurement in the cheek, continuous measurement) measured by Tewameter® TM300 (Courage + Khazaka, electronic GmbH, Cologne, Germany) at D15, D30 and D60;
- skin elasticity (1 replicate measurement in the cheek) measured by Cutometer® MPA 580 (Courage + Khazaka, electronic GmbH, Cologne, Germany) (on time 2.0 s, off time 2.0 s, pressure 450 mbar, repetition 3, total time 12 s) at D15, D30 and D60;
- skin surface properties (smoothness, roughness and wrinkledness) (1 replicate measurement in the periocular area) measured by Visioscan® VC 98 (Courage + Khazaka, electronic GmbH, Cologne, Germany) at D15, D30 and D60;
- skin dermatological control by a board-certified dermatologist at D15, D30 and D60.

All of the measurements were carried out on cleansed faces under temperature- ($21 \pm 1$ °C) and humidity ($50 \pm 10\%$) controlled conditions. For 15 to 20 min before the beginning of the physical measurements, subjects were left to acclimate to the room conditions.

Volunteers' perception of product efficacy on skin following parameters: moisturizing level, skin elasticity, pulling sensation, desquamation, softness and smoothness of the skin evaluated by a self-assessment questionnaire at D60.

*2.5. Sample Size*

Sample size was calculated with a two-sided 5% significance level ($\alpha$) and a power (1-$\beta$) of 80% taking into account the standard deviation and the variation obtained for skin hydration in a previous pilot study (data not shown). The statistical model was the one-sided one-sample *t*-test. According to the statistical model, a sample size of 13 subjects per group achieves 100% power to detect the variation of hydration seen in the pilot study. Seven additional volunteers were recruited in each group in case of possible dropout during the intervention period.

*2.6. Randomization and Blindness*

After the enrollment, subjects were randomly assigned to one of the three study groups, in a 1:1:1 ratio, to receive active products or a placebo. For allocation, a computer-generated (using PASS 11 statistical software, version 11.0.8 for Windows; PASS, LLC, Kaysville, UT, USA) restricted randomization list (Wei's urn algorithm) was used.

Subjects, investigator, and collaborators were kept blind to group assignment.

*2.7. Statistical Methods*

Statistical analysis was carried out on the intention-to-treat (ITT) population using NCSS 8 (version 8.0.4 for Windows; NCCS, LLC) running on a Windows 2008 R2 64 Edition server (Microsoft, Redmond, WA, USA). Data normality (both for raw data and variations vs. the baseline value) was verified using the Shapiro–Wilk W normality test and data shape. Comparisons were carried out to assess if the treatments (between-subjects) were source of variation of the measured endpoints. Data (differences adjusted for baseline) were submitted to RM-ANOVA followed by Tukey–Kramer Multiple-Comparisons post-test. The statistical significance was set at $p < 0.05$. $p$-values are reported as follows: ***, $p < 0.001$; **, $p < 0.01$; and *, $p < 0.05$. Results are reported as mean $\pm$ SD (Standard Deviation).

## 3. Results

*3.1. Participant Flow and Recruitment*

Subjects attended clinic visits at the time of randomization (baseline) and after 15, 30, and 60 days of product use. All of the randomized subjects ($n = 20$ per group) completed the study (Figure 2).

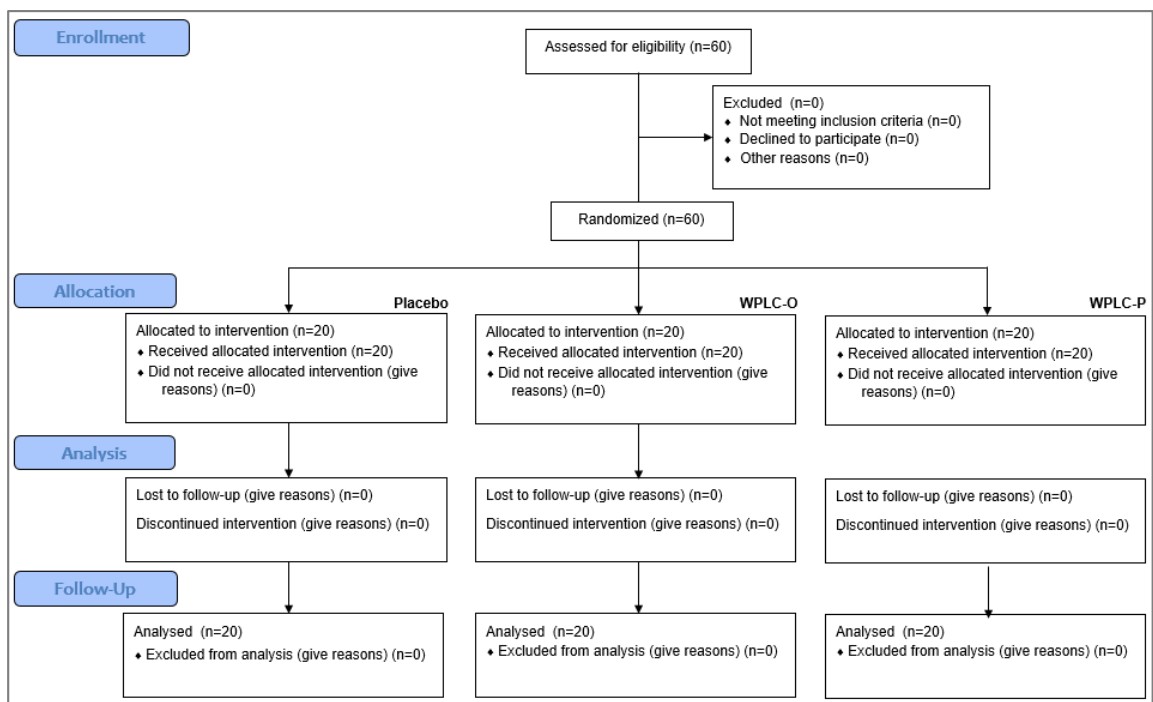

**Figure 2.** Study participant flow chart.

In accordance with the protocol requirement, the volunteers did not change their dietary habits during the study (data not shown).

The tested products were well tolerated overall by all the volunteers throughout the study. Self-assessment questionnaire results demonstrated that tolerance was found to be "fairly good" and "excellent" by 95%, 90% and 85% of the volunteers respectively supplemented with placebo, WPLC-P and WPLC-O products.

*3.2. Baseline Data*

Subjects' baseline demographics and clinical characteristics are presented in Table 5 for each group. During the baseline visit skin hydration was measured from Monday to Friday in order to obtain a consolidate value.

**Table 5.** Subjects' baseline demographic and clinical characteristics.

| Baseline characteristics | Placebo | WPLC-O | WPLC-P |
|---|---|---|---|
| Number of subjects | 20 | 20 | 20 |
| ge (years) | $48.3 \pm 8.6$ | $44.3 \pm 8.7$ | $45.7 \pm 9.7$ |
| Weight( kg) | $60.7 \pm 10.1$ | $59.0 \pm 10.3$ | $58.1 \pm 6.6$ |
| BMI (kg/m$^2$) | $22.9 \pm 3.2$ | $22.5 \pm 3.2$ | $22.2 \pm 2.9$ |
| Menopause | 8 (40%) | 5 (25%) | 6 (30%) |
| Smokers | 5 (25%) | 8 (40%) | 8 (40%) |
| Cigarettes (n/day) | $11.4 \pm 7.5$ | $12.4 \pm 8.9$ | $11.0 \pm 6.6$ |
| Skin hydration (c.u) | $44.5 \pm 4.5$ | $44.7 \pm 5.0$ | $43.3 \pm 5.6$ |
| TEWL (g/h/m$^2$) | $10.9 \pm 2.7$ | $11.3 \pm 3.1$ | $11.6 \pm 3.3$ |
| Skin elasticity (Ratio Ua/Uf) | $0.564 \pm 0.064$ | $0.581 \pm 0.066$ | $0.570 \pm 0.074$ |
| Skin smoothness (SEsm (a.u.)) | $38.97 \pm 5.88$ | $38.86 \pm 5.02$ | $39.16 \pm 5.61$ |
| Skin roughness( SEr (a.u.)) | $1.99 \pm 0.37$ | $1.90 \pm 0.39$ | $1.91 \pm 0.36$ |
| Skin wrinkledness( SEw (a.u.)) | $35.37 \pm 3.52$ | $35.66 \pm 3.98$ | $34.89 \pm 2.54$ |

Data are averages (±standard deviation) or number of subjects (%).

*3.3. Skin Hydration and Skin Barrier Function*

Skin hydration values measured by Corneometer® are reported in Table 6.

Skin hydration was significantly increased ($p \leq 0.001$) after 15, 30 and 60 days of supplementation with WPLC-P and WPLC-O extracts compared to baseline (D0). On the contrary, in the placebo group, skin hydration was only increased after 30- and 60-day supplementation periods ($p < 0.05$ day 0). Variations in skin hydration measured at D15, D30 and D60 for both the WPLC-O and the WPLC-P groups were statistically different ($p < 0.001$) compared to placebo. There were no significant differences at each treatment time between the two active groups.

As reported in Table 6, similar improvements were obtained for TEWL. TEWL was significantly decreased ($p \leq 0.001$) after 15, 30 and 60 days of supplementation with WPLC-P and WPLC-O extracts compared to baseline. On the contrary, TEWL remained unchanged in the placebo group during the 60 days supplementation period ($p > 0.05$ vs. baseline and day 0). Variations in TEWL measured at D15, D30 and D60 for both the WPLC-O and the WPLC-P groups were statistically different ($p < 0.001$) compared to placebo. There were no significant differences at each treatment time between the two active groups.

**Table 6.** Skin Hydration and TEWL results after 15, 30 and 60 days of supplementation with WPLC-O and WPLC-P extracts measured by Corneometry® and Tewameter®. Data are averages ± SD in corneometric units (c.u.) and in TEWL (g/h/m$^2$).

| Time (days) | Corneometer® (c.u) | | | TEWL (g/h/m$^2$) | | |
|---|---|---|---|---|---|---|
| | Placebo | WPLC-O | WPLC-P | Placebo | WPLC-O | WPLC-P |
| D0 | 44.5 ± 4.5 [a] | 44.7 ± 5.0 [a] | 43.3 ± 5.6 [a] | 10.9 ± 2.7 [b] | 11.3 ± 3.1 [b] | 11.6 ± 3.3 [b] |
| D15 | 46.2 ± 6.1 [a] (+3.6%) | 54.8 ± 6.5 [b] (+23.4% [†]) | 53.6 ±7.0 [b] (+24.9% [†]) | 11.2 ± 2.0 [b] (+5.3%) | 9.4 ± 2.4 [a] (−16.2% [†]) | 9.7 ± 3.9 [a] (−16.0% [†]) |
| D30 | 47.8 ± 5.8 [b] (+7.5 %) | 56.3 ± 6.0 [c] (+26.7% [†]) | 55.4 ±5.8 [c] (+29.4% [†]) | 10.8 ± 2.8 [b] (+1.0%) | 9.1 ± 2.5 [a] (−18.3% [†]) | 8.8 ± 2.9 [a] (−22.6% [†]) |
| D60 | 48.1 ± 6.5 [b] (+8.2%) | 60.9 ± 7.0 [d] (+37.2% [†]) | 58.6 ±10.2 [d] (+36.8% [†]) | 10.9 ± 3.0 [b] (+3.8%) | 9.0 ± 2.5 [a] (−19.5% [†]) | 9.3 ± 2.6 [a] (−19.0% [†]) |

In brackets is reported the % variation vs. D0; NS: Not significant; Significantly different from Day 0: a < b < c < d, $p < 0.05$; † Significantly different ($p < 0.001$) from the placebo group.

*3.4. Skin Elasticity*

Skin elasticity values measured by Cutometer® are reported in Table 7.

**Table 7.** Skin elasticity results after 15, 30 and 60 days of supplementation with WPLC-O and WPLC-P measured by Cutometer®. Data are averages $\pm$ SD in elasticity values R2 (Ua/Uf).

| Time (days) | Skin elasticity (R2 (Ua/Uf)) | | |
|:---:|:---:|:---:|:---:|
| | **Placebo** | **WPLC-O** | **WPLC-P** |
| **D0** | 0.564 $\pm$ 0.064 [a] | 0.581 $\pm$ 0.066 [a] | 0.570 $\pm$ 0.074 [a] |
| **D15** | 0.558 $\pm$ 0.067 [a] ($-$0.7%) | 0.665 $\pm$ 0.083 [b] (+14.9% [†]) | 0.661 $\pm$ 0.064 [b] (+16.6% [†]) |
| **D30** | 0.589 $\pm$ 0.083 [a] (+5.2%) | 0.698 $\pm$ 0.072 [b] (+20.9% [†]) | 0.697 $\pm$ 0.060 [b] (+23.2% [†]) |
| **D60** | 0.606 $\pm$ 0.070 [a] (+8.3%) | 0.788 $\pm$ 0.085 [c] (+36.6% [†]) | 0.766 $\pm$ 0.076 [c] (+35.9% [†]) |

In brackets is reported the % variation vs. D0; NS: Not significant; Significantly different from Day 0: a < b < c, $p < 0.05$; † Significantly different ($p < 0.001$) from the placebo group.

Skin elasticity was significantly increased ($p \leq 0.001$) after 15, 30 and 60 days of supplementation with WPLC-P and WPLC-O extracts compared to baseline (D0). On the contrary, it remained unchanged in the placebo group during the 60 days supplementation period ($p > 0.05$ vs baseline). Variation of skin elasticity in the WPLC-O and the WPLC-P groups was statistically different ($p < 0.001$) from the placebo group after 15, 30 and 60 days of supplementation. There were no significant differences at each treatment time between the two active groups.

*3.5. Skin Aging Parameters*

Smoothness, roughness and wrinkledness data are reported in Table 8 and in Table 9.

Skin roughness and wrinkledness were significantly decreased while smoothness was significantly increased after 15, 30 and 60 days of supplementation with WPLC-P and WPLC-O extracts compared to baseline (respectively $p \leq 0.001$, $p \leq 0.001$ and $p \leq 0.001$ in both groups). In the placebo group, skin roughness was significantly decreased after 60 days of supplementation while smoothness was significantly increased after 30 and 60 days of supplementation compared to baseline. However, variations in skin aging parameters measured at D15, D30 and D60 for both the WPLC-O and the WPLC-P groups were statistically different ($p < 0.001$, $p < 0.001$ and $p < 0.001$, respectively) compared to placebo. On the contrary, there were no significant differences at each treatment time between the two active groups.

**Table 8.** Skin smoothness and roughness values after 15, 30 and 60 days of supplementation with WPLC-O and WPLC-P extracts. Data are averages ± SD in SESm (a.u.) and SEr (a.u.) values.

| Time (days) | Smoothness (a.u.) | | | Roughness (a.u.) | | |
|---|---|---|---|---|---|---|
| | Placebo | WPLC-O | WPLC-P | Placebo | WPLC-O | WPLC-P |
| D0 | 38.97 ± 5.88 [a] | 38.86 ± 5.02 [a] | 39.16 ± 5.61 [a] | 1.99 ± 0.37 [c] | 1.90 ± 0.39 [c] | 1.91 ± 0.36 [c] |
| D15 | 39.39 ± 5.40 [a] (+0.43%) | 43.14 ± 4.18 [b] (+4.28% [∫]) | 43.48 ± 5.72 [b] (+4.32% [∫]) | 1.92 ± 0.62 [c] (−0.08%) | 1.28 [b] ± 0.51 (−0.62% [†]) | 1.32 ± 0.47 [b] (−0.59% [†]) |
| D30 | 40.87 ± 5.26 [b] (+1.90%) | 48.33 ± 5.91 [c] (+9.47% [†]) | 49.42 ± 5.58 [c] (+10.26% [†]) | 1.77 ± 0.69 [c] (−0.22%) | 0.87 [a] ± 0.35 (−1.03% [†]) | 0.84 ± 0.40 [a] (−1.06% [†]) |
| D60 | 41.29 ± 5.54 [b] (+2.33%) | 52.97 ± 5.11 [d] (+13.81% [†]) | 52.33 ± 5.14 [c] (+13.18% [†]) | 1.63 ± 0.61 [b] (−0.36%) | 0.69 [a] ± 0.28 (−1.20% [†]) | 0.68 ± 0.37 [a] (−1.23% [†]) |

In brackets is reported the % variation vs. D0; NS: Not significant; Significantly different from Day 0: a < b < c < d, $p < 0.05$; † ($p < 0.001$) and ∫ ($p < 0.01$), significantly different from the placebo group.

**Table 9.** Skin wrinkledness variation results after 15, 30 and 60 days of supplementation with WPLC-O and WPLC-P. Data are mean ± SD in SEw (a.u.).

| Time (days) | Skin Wrinkledness (SEw (a.u.)) | | |
| --- | --- | --- | --- |
| | Placebo | WPLC-O | WPLC-P |
| D0 | 35.37 ± 3.52 [d] | 35.66 ± 3.98 [d] | 34.89 ± 2.54 [d] |
| D15 | 36.11 ± 3.04 [d] (+0.74%) | 32.56 ± 2.28 [c] (−3.10% [†]) | 32.14 ± 2.13 [c] (−2.75% [†]) |
| D30 | 35.81 ± 3.36 [d] (+0.44%) | 30.68 ± 2.20 [b] (−4.97% [†]) | 30.57 ± 2.50 [b] (−4.33% [†]) |
| D60 | 35.01 ± 4.14 [d] (−0.36%) | 28.87 ± 3.40 [a] (−6.79% [†]) | 28.60 ± 2.90 [a] (−6.29% [†]) |

In brackets is reported the % variation vs. D0; NS: Not significant; Significantly different from Day 0: a < b < c < d, $p < 0.05$; † ($p < 0.001$), significantly different from the placebo group.

### 3.6. Self-assessment Questionnaire Results

At day 60, self-assessment questionnaire results demonstrated that volunteers own perceptions of product efficacy are consistent with the measures performed on their skin. Results of the questionnaire are presented in Table 10.

**Table 10.** Results of the self-assessment questionnaire after 60 days of supplementation with WPLC-O and WPLC-P.

| Questions and Self-Assessment | Placebo | WPLC-O | WPLC-P |
| --- | --- | --- | --- |
| **How do you evaluate the food supplement efficacy concerning skin moisturizing level improvement?** | | | |
| Insufficient | 30% | 5% | 5% |
| Sufficient | 50% | 30% | 35% |
| Fairly good | 15% | 45% | 35% |
| Excellent | 5% | 20% | 25% |
| **How do you evaluate the food supplement efficacy concerning skin elasticity?** | | | |
| Insufficient | 45% | 5% | 5% |
| Sufficient | 45% | 30% | 35% |
| Fairly good | 5% | 45% | 35% |
| Excellent | 5% | 25% | 25% |
| **Do you think that your skin presents less "pulling sensation"?** | | | |
| Yes | 25% | 80% | 75% |
| No | 75% | 20% | 25% |
| No opinion | 0% | 0% | 0% |
| **Do you think that your skin presents less desquamated?** | | | |
| Yes | 20% | 85% | 80% |
| No | 75% | 15% | 15% |
| No opinion | 5% | 0% | 5% |
| **Do you think that your skin is softer and smoother?** | | | |
| Yes | 25% | 85% | 90% |
| No | 65% | 15% | 10% |
| No opinion | 10% | 0% | 0% |
| **Do you think that the aspect of your skin looks better?** | | | |
| Yes | 15% | 80% | 80% |
| No | 65% | 15% | 20% |
| No opinion | 20% | 5% | 0% |

## 4. Discussion

The purpose of this double-blind randomized controlled human clinical trial was to test, in a rigorous way, the hypothesis that even small amounts of sphingoid base derivatives, administered orally to humans, can afford measurable and significant and perceivable benefits to the skin,

as suggested by the various animal studies and a few smaller-scale human trials [9–17] and our own preliminary study.

For this purpose, a specific purified wheat flour extract composed of wheat polar lipids rich in GluCers and DGDG was extracted, in both a powder and an oil based form, and formulated in capsules with identical aspect for oral intake over two months by healthy human volunteers presenting with dry skin and clinical skin aging signs. Placebo capsules were provided to a control group. Even if the level of active ingredient was similar, we tested both powder and oil form of WPLC to verify whether the galenic form had an impact or not on product efficacy.

In both treated groups, oral intake supplementation of 30 mg/day of WPLC-P or 70 mg/day of WPLC-O, providing an average of 1.7 mg of glucosylceramides and 11.5 mg of DGDG, induced a strong and highly significant improvement in skin hydration markers and skin anti-aging effects compared to the placebo group. As expected, given the almost identical content in the active components, DGDG and GluCers, there were no significant differences of efficacy between the two WPLC groups (powder form and oil form). Moreover, measures performed after 15, 30, and 60 days of supplementation showed that efficacy on each studied parameter increased significantly with time compared to the baseline. It is interesting to note that these significant effects appeared at the very early stage of the supplementation, in only 15 days and continued over the entire period of the study. With respect to the clinical pertinence of the statistically significant observation: while we did not include subjective "dermatological" scoring of the skin care effects, the self-assessment answers from the panelists in each group clearly showed strongly perceived benefits and therefore preferences for the verum capsules as opposed to the placebo. In view of the reported regulatory of food habits and intake, this factor cannot be considered as a source of data variation during the study period.

Like essential fatty acids, sphingosines and sphinganines, the sphingoid base structures in SC ceramides, cannot be synthetized de novo by the human organism. They are generated by degradation of extracutaneous lipids coming from the general diet [9,10,12].

The fate of ingested sphingolipids in human beings remains to be understood by further studies. Based on the present state of knowledge, we can reasonably assume that sphingolipid metabolism in humans will follow a similar path of digestion and absorption to that described in animal studies, producing sphingolipid metabolites that are absorbed intestinally and distributed to blood. Based on this hypothesis, metabolites of dietary sphingolipids, including ceramides and glucosylceramides, could reach the skin, enter the metabolic processes in the epidermis through the Golgi apparatus and lamellar bodies, and thus participate in the de novo synthesis of ceramides and other physiological skin lipids in situ. Consequently, orally taken sphingolipids could play their functional role in the stratum corneum to maintain or restore dry skin conditions and improve skin microrelief.

Cereals, particularly barley, corn, and wheat, are DGDG-rich [27]. In rats orally fed with DGDG, the fatty acids generated after hydrolysis were shown to be reesterified into triglycerides and phospholipids in the enterocytes and then transported by triglyceride-rich lipoproteins [28].

The presence of DGDG in WPLC is likely to improve the efficacy of GluCers by increasing intestinal absorption. For instance, as shown in a human clinical study [29], oral supplementation of phytosterol dispersed in a diacylglycerol-rich oil increased the health benefits of phytosterols compared to a triacylglycerol oil. The authors explained that the reason for this could be that diacylglycerol-rich oil is a better solvent for hydrophobic components as phytosterols. More recently, it was demonstrated that bioaccessibility of beta-carotenoids was markedly enhanced when solubilized in diacylglycerol (dioleate) before digestion compared to beta-carotene alone. The low bioavailability of lipophilic micronutrients is mainly caused by their limited solubilization into aqueous micelles, which hinders their ability to be taken up by the intestine [30].

DGDG is in fact a specific diacylglycerol where two galactoses are esterified, with high emulsifying properties [31]. By analogy with diacyl glycerol oil, DGDG could be largely responsible for an increase in the distribution of dietary sphingolipids including GluCERs in bile acid micelles after ingestion, followed by improvement of intestinal absorption of both intact forms and metabolites.

Thanks to the amphiphilic structure of DGDG, sphingolipids, including GluCERs that composed WPLC, can be dispersed in highly stable micellar dispersion in water.

Taking all these observations and the present study data into account, it appears reasonable to correlate the demonstrated improvements in age-related symptoms such as skin hydration, elasticity and smoothness with the single-parameter of WPLC supplementation in this human clinical trial.

## 5. Conclusions

In a human clinical trial based on a gold standard study design, we show that daily oral supplementation of purified wheat glucosylceramides and DGDG induced a strong and highly significant improvement in skin hydration markers compared to placebo, after only 15 days and beyond.

The level of efficacy on skin hydration and TEWL was higher and was detected faster than data reported in other human clinical studies on sphingolipids or ceramides oral supplementations. For the first time, we demonstrated the anti-aging properties induced by wheat sphingolipids, including GluCers and DGDG, issued from WPLC oral supplementation, on four skin markers concomitantly: elasticity, smoothness, roughness, and wrinkledness. Purified WPLC can be considered as an interesting food supplement ingredient inducing healthy and young-looking skin.

Although the clinical research has reached its aim, there were some unavoidable limitations. Indeed, interactions between the skin and the basal cream were possible and cannot be rejected a priori. Some of the possible interaction also takes into account the "placebo" effect. However, even if some interactions occurred, we think that they do not represent a limitation of study results interpretation. In fact, the level of skin parameter improvements obtained in the active treatment arms is significantly higher than the one measured in the placebo arm. In addition, some inflation of the overall alpha-error can occur due to the number of statistical tests. Further studies to investigate the bioavailability of wheat sphingolipids and DGDG metabolites after digestion, their effectiveness on ceramides metabolism pathway in the skin, the enhancement of ceramides content at the skin level, and the specificity of ceramide structures and associated lipids on their efficacy are planned.

**Acknowledgments:** We sincerely express our gratitude for valuable advice and comments to Claire Notin from SEPPIC SA, 22 Terrasse Bellini, Paris la Défense, Puteaux, France.

**Author Contributions:** V.M.B. conceived and designed the experiments; E.C. and A.M. performed the experiments; V.N. analyzed the data; V.M.B. wrote the paper.

**Conflicts of Interest:** The authors declare no conflict of interest.

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
