# Peer review of "Improving Skin Hydration and Age-related Symptoms by Oral Administration of Wheat Glucosylceramides and Digalactosyl Diglycerides: A Human Clinical Study"

_cosmetics, doi:10.3390/cosmetics4040037_

Round 1

Reviewer 1 Report

Thank you very much for inviting me to review this manuscript. Please find my comments below:

(1) Intro, page 1, second para: Please provide reference for the 30% figure.

(2) Intro, page 2, fourth para: Please be a little bit more precise here, what exactly was improved. I assume the skin became smother?

(3) Intro, page 2, last para: I assume you are referring to preliminary data. What are “healthy dry and wrinkled skin human volunteers”? Delete the last sentence. Just clearly state your research question.

(4) Methods: I strongly recommend to restructure this section exactly according to CONSORT 2010. At the moment it is difficult to follow. The composition of the oral supplementation can be described under item 5.

(5) Methods, page 4: Please use past tense in this section. Please provide reference that a Corneometer reading below 50 corresponds to dry skin. What are ‘medium’ photoageing sings. Please list in- and exclusion criteria completely and clearly (CONSORT item 4). What is mild to moderate according to Fitzpatrick? This is a phototype classification.

(6) Methods, page 5: Why was a cream given prior and during the study. Every leave-on products has an effect on skin structure and function. This was most probably measured during the trial.

(7) Outcomes: Please state the investigational skin areas. Please describe exactly how the measurements were done: replicate measurements? On-off-cycles of the cutometer (suction pressure? Probe opening?) etc…

(8) Questionnaire: Please describe the response options.

(9) Sample size: Please state clearly you’re a priori assumptions including assumed delta. If you use a one sided t-test for sample size estimation you must use this test for analyzing the primary outcome. At the moment there is a mismatch between this and subsequent analysis. A t-test is not mentioned in 2.8 and it also makes no sense because you compare three groups.

(10): Results, clinical data, page 8: The reference to Table 4 makes no sense. Do not compare baseline parameters statistically in an RCT (see CONSORT). Figures in the grouped bar graphs and tables are identical. I recommend to provide tables only. Consider the SAMPL guidelines for statistical reporting, e.g. p-values must have no more than three decimal places, provides SD for estimates but SE for mean differences etc. Overall there are so many statistical tests increasing the overall alpha-error. Please adjust this error probability and/or reduce the number of tests.

(11) Discussion, first para: You are talking about ‘significant’ benefits. Do you mean ‘clinically relevant’? How are the parameters related to clinical relevance? Is there any support that an increase of SCH of 5 is beneficial?

(12) Discussion, fourth para: You did a statistical analysis of food intake? Please discuss only results which have been presented before. Same is true for self-perception results in the next para.

(13) Please add study limitations.

 (14) Conclusion: what are “anti-ageing properties”? If you use this concept in the Conclusion please introduce it before in the Intro.

Author Response

Dear Madam, dear Sir,

Please find attached our reply to your comments.

Best regards,

Valérie BIZOT

Reviewer 2 Report

This paper describes a human trial, carried out in double blind vehicle controlled randomized manner, to study the skin beneficial effects obtained by oral absorption of "ceramide" containing capsules, over a period of 60 days. The reported results are rather impressive.

The number of research papers on such "oral cosmetics" (a regulatory oxymoron) is slowly increasing, and both animal tests as well as some human trials seem to converge towards credible benefits of such nutritional supplementation. It is still surprising that such small amounts of these additional ceramides and glycosylceramides can achieve so much, when the average intake of ceramide-type substances in "normal" food is so much higher. Clearly, questions of gastrointestinal bioavailability must play an important role. The data on DGDG in this paper do hint at this hypothesis, although they cannot verify it in detail. It is therefor a bit regrettable that in this present study two capsules with too closely related composition were investigated side by side. A future study with the same ceramide extract, but devoid of DGDG, might contribute to better understanding. If possible, a future study protocol should try to include measuring and quantifying the amount of ceramides in the skin, on D0 and on D60. Would we see an increase to further strengthen the argument that the orally taken ceramides do help the skin to make more on its own?

The analytical data on the extracts are useful and a refreshing change from so many undefined extracts being tested and reported. 

Minor details:

the paper contains 11 tables and 7 figures. But much of the figures and the corresponding table report the same data, just in a different way. Given that the precise percentages of changes in these parameters are not physical constants (such as wavelengths, frequencies, chemical shifts etc. which, from one experiment to another, are strictly reproducible), but are  individually widely variable values, averaged out over the panel but certainly not reproduced as such if the study were run a second time, it does not seem necessary to include all these tables the data of which are redundant with the corresponding graphs (one or two might suffice for illustrative purposes). The graphs should be redesigned in such a way as to indicate the significance of each change versus the D0 AND versus the placebo. Although the legends say that the colored asterisks correspond to DX vs D0, the graphs do not show this (no lines from D0 to D15/30/60) .Too many stars and not enough connectors...

On  page 2 there is mention of a reference [11], but there are two references [10] in the bibliography. Please correct.

In the last paragraph of the Discussion (line 398) I object to the term "anti-ageing" effects; the term is popular in a commercial context, but scientifically it has no sufficiently precise meaning. I suggest to write "to correlate the demonstrated improvements in age related symptoms such as skin hydration, elasticity and smoothness with the single parameter..."

Author Response

Dear Madam, dear Sir,

You will find attached our reply to your different comments on our manuscript.

Best regards,

Valerie BIZOT

Round 2

Reviewer 1 Report

Thank you very much for reviewing the revised manuscript again. The structure has improved substantially. From my point of view there are some points that need to be addressed:

(1)    Sample size: In the response is mentioned, that priori sample size calculations are based on statistical tests that are later not used. The opposite is the case. You must exactly conduct the statistical analysis as planned before because it is a summed model for this study. Because this has not been done here please describe this transparently. Consequently this was not a confirmatory study. Please adjust your discussion and interpretation accordingly.

(2)    Again, do not compare baseline values with a statistical test. Remove this from the text. A test does not indicate an “unbiased randomization” or the “absence of covariates”.

(3)    Results: Please correct carefully the verbs describing the changes. I think TEWL, roughness etc. decreased in the intervention groups…

(4)    Regarding on products: I totally agree that you cannot remove leave-on products during a trial in subjects with dry skin. However, now there is a problem now in the eligibility criteria. Here you say that subjects were excluded when “using topical products containing moisturizing … ingredients”. This is a contradiction. Does this mean that all subjects started treating their skin when starting the trial? This makes no sense.

(5)    Please add limitations: Having 20 subjects per group is no limitation per se. Limitations are that is was not confirmatory, that there might be an interaction between the topical and the oral products, you have done a high number of statistical tests inflating the overall alpha error etc… Please say this in the text.

(6)    Results: Please provide SDs as spread estimates around your numbers. SEMs are only appropriate indicating mean differences (see CONSORT).

Author Response

Dear Sir,

You will find attached the cover letter with the details of the modifications made according your comments.

Yours faithfully

Dr Valérie BIZOT
